# Challenges of the Immunotherapy: Perspectives and Limitations of the Immune Checkpoint Inhibitor Treatment

**DOI:** 10.3390/ijms23052847

**Published:** 2022-03-05

**Authors:** Paula Dobosz, Maria Stępień, Anna Golke, Tomasz Dzieciątkowski

**Affiliations:** 1Department of Genetics and Genomics, Central Clinical Hospital of the Ministry of Interior Affairs and Administration in Warsaw, 02-507 Warsaw, Poland; 2Department of Infectious Diseases, Medical University of Lublin, 20-059 Lublin, Poland; mmaria.stepien@gmail.com; 3Department of Sports Medicine, Medical University of Lublin, 20-059 Lublin, Poland; 4Department of Preclinical Sciences, Institute of Veterinary Medicine, Warsaw University of Life Sciences, 02-776 Warsaw, Poland; anna_golke@interia.eu; 5Chair and Department of Medical Microbiology, Medical University of Warsaw, 02-005 Warsaw, Poland

**Keywords:** checkpoint inhibitors, immunotherapy, cancer, tumour microenvironment, immune system

## Abstract

Immunotherapy is a quickly developing type of treatment and the future of therapy in oncology. This paper is a review of recent findings in the field of immunotherapy with an emphasis on immune checkpoint inhibitors. The challenges that immunotherapy might face in near future, such as primary and acquired resistance and the irAEs, are described in this article, as well as the perspectives such as identification of environmental modifiers of immunity and development of anti-cancer vaccines and combined therapies. There are multiple factors that may be responsible for immunoresistance, such as genomic factors, factors related to the immune system cells or to the cancer microenvironment, factors emerging from the host cells, as well as other factors such as advanced age, biological sex, diet, many hormones, existing comorbidities, and the gut microbiome.

## 1. Introduction

Despite several spectacular and successful immunotherapy trials of recent years, many cancer patients do not respond to immunotherapy, or their response remains short-lived, and the recurrence of the disease seems to be inevitable mainly due to the rapidly evolving resistance. In fact, it is the resistance itself, either primary or acquired, that remains poorly understood; thus, it is impossible to overcome without further detailed research. Undoubtedly, we should search for the answer on the molecular level, since the mechanisms underlying resistance are clearly genetic- and/or protein-related. As we have already learned from the complex and multi-faceted structure of the immunological synapse, the mechanisms of the resistance to the immune checkpoint inhibitors seem to be even more complicated and multifactorial, possibly engaging such elements as gene expression, cellular metabolism, the presence or absence of inflammation, angiogenesis, or tumour neovascularization [1]. In this article we try to cover the latest knowledge regarding the resistance to the immune checkpoint inhibitor therapy. Even if this type of bioactive molecule represents only a small portion of existing immunotherapies, many already described factors contributing to the development of resistance are common for different types of immunotherapy and beyond, including CAR T cells or even oncolytic viruses. It is, however, worth nothing that the following list might be just an introductory overview or an invitation to further investigations.

## 2. Primary versus Acquired Resistance to Immunotherapy

Before we enter a discussion of the actual factors contributing to immunotherapy resistance, it might be important to highlight the difference between those two types of resistance: primary and acquired. Primary resistance to immunotherapy—sometimes known as primary insensitivity to immunotherapy—occurs when cancer does not respond to the treatment which has never been used in this patient before. In most cases the immune system does not take any actions despite the pharmacological stimulation used, as it may happen in the case of the immune checkpoint inhibitor drugs [2,3]. On the other hand, acquired immunotherapy resistance occurs when the drug used before no longer works, despite being successfully used in the past in particular patients; at least 6 months of progression-free survival must be observed to meet the criteria of acquired resistance [1,2,3]. To exemplify this phenomenon, among melanoma patients treated with immune checkpoint inhibitors, about 30% of them do respond well at the beginning of the treatment but develop acquired resistance during the treatment regime up to the point that they finally stop responding to this form of immunotherapy [4,5].

According to the binding criteria of the American Society for Immunotherapy of Cancer, primary resistance to immune checkpoint inhibitors needs to meet at least three requirements: exposure to the drug for at least six weeks, stable disease, or progressive disease as a response for at least six months, confirmatory scan for progressive disease after at least four weeks of the initial progression [6]. Immunotherapy cannot launch an antitumour immune response or tumour-induced immunosuppression cannot be tempered as a result of existing immunotherapy resistance. Investigating the mechanisms of the primary and acquired resistance to checkpoint inhibitors has become one of the most important challenges in the field of cancer immunotherapy.

Nevertheless, it should not be surprising to discover that similar factors might be responsible for both forms of the resistance phenomenon. Several mechanisms have already been described, whereas others remain putative and require deeper exploration. Known molecular mechanisms of resistance include the lack of antigen expression, constitutive PD-L1 expression or other co-inhibitory molecules, EGF/EGFR expression abnormalities, loss of HLA expression or presentation, improper activation of the MAPK pathway, WNT/β-catenin pathway activation, loss of the PTEN expression leading to the enhancement of the PI3K pathway, and abrupted interferon pathway signalling, especially INFγ, JAK/STAT signalling pathway abnormalities [6]. Arguably during the first stages of the tumour growth and development there seem to be less factors contributing to resistance, whereas progressing cancer and its incredibly complex microenvironment comprise of an entire plethora of factors possibly rendering immunotherapy resistance. This obviously challenges the treatment success irrespective of the type of therapy and contributes to the usual failure in advanced cancer treatment. Given the complicated interplay between all of the factors of the tumour entity itself, it might even be difficult to unambiguously classify each factor of the tumour microenvironment as belonging to the tumour cells products and derivatives or the actions of the immune system itself. In fact, one protein can be produced either by tumour cells or the immune cells, or even both. For the clarity of the article, presented contributing factors have been classified explicitly, although one should remember that the tumour microenvironment might be far more complex and involute. Factors described in this article have been presented in a simplified way in Figure 1. A comparison of primary vs. acquired resistance to immunotherapy was collated in Table 1. 

## 3. Genomic Factors and Tumour Heterogeneity

Low immunotherapy response rates in case of the most prevalent cancer types, such as breast, prostate or pancreas cancers, force scientists to search for new therapeutic solutions, but even more importantly to search for the molecular mechanisms underlying those phenomena. It might be the solid tumour’s heterogeneity that impacts the diversified response to the same treatment [9]. Surely, it is for the tumour internal heterogeneity, comprising many different clones, but also metastasis heterogeneity, since every single secondary tumour may pose its own unique genetic changes [1,9,10]. Moreover, this is never a constant situation, given the fact that every cancer entity evolves. One may also observe that only a fraction of metastasis responds to treatment, whereas others remain intact [1,9]. This may suggest the presence of different genetic variants in cancer cell clones giving rise to the new metastasis. There are already several examples of such situations being observed in the clinic, for example brain metastases of the non-small-cell lung carcinoma (NSCLC). In this case, newly formed metastases are usually highly heterogeneous and show very different expression patterns than the primary tumour [11]. In the case of PD-L1 expression, it has been noted that metastases present within the chest and around the primary tumour may have high expression rates, whereas distant metastases do not express PD-L1 at all, such as in the case of the brain secondary tumours rendering them insensitive to checkpoint inhibitor immunotherapy [11]. Luckily, these extreme situations are rather unique, and generally the expression of immune checkpoint genes is relatively stable within the same patient’s tumour and its metastases [12]. 

Tumour mutation burden, TMB, is a parameter initially considered as a potential biomarker of the immunotherapy response [1,13]. The correlation between high TMB and the response to immunotherapy with the use of anti-CTLA-4 antibodies was significant among advanced melanoma patients, whereas the correlation between high TMB and the anti-PD-1 immunotherapy was significant among NSCLC, to give some examples [14,15,16,17]. Therefore, TMB has been included in several immunotherapy recommendations as an integral part of diagnostic proceedings and is still being used. However, more accurate studies available today cause us to rethink this approach and undeniably to verify TMB as an immunotherapy biomarker indicating whether or not to administer potentially life-saving drugs to our patient [13]. It is clear that even cancers having very similar TMB can respond differently to the same medication, such as in the case of the anti-PD-1 antibodies, consolidating the notion of cancer multifactorial complexity and imponderability [13].

Similarly, microsatellite instability and DNA repair mechanism deficiency, such as mismatch repair deficiency, have already been linked to the immunotherapy response [1,18]. In several cancer types, such as gastroesophageal adenocarcinoma, subsets with microsatellite instability and EBV positive cases, positive response to the immune checkpoint inhibitors has been observed, whereas the same cancer type but chromosomal stable and genome stable subtypes have a significantly lower response to this type of drug [6]. Later studies revealed that gastroesophageal adenocarcinoma non-responding subtypes have T cell exclusion as a main mechanism of immunosuppression, but not T cell suppression as such [6]. All of the factors mentioned above are related to the higher probability of neoantigens presence on the tumour cell membranes being tumour-specific, thus raising the tumour’s immunogenicity and possible T cell infiltration [16,19,20]. Even if theoretically positive, those effects might not necessarily correlate with better response to immunotherapy, since cancer cells can inhibit immune system actions, for example through cytokine release or surface protein expression causing T cells anergy and exhaustion [21]. Also, cancer cells might be resistant to cellular stress and autophagy pathways, as well as to many further factors potentially contributing to immunotherapy resistance [22].

Genetic variants impacting crucial cell signalling pathways invariably create the basis for the development of the anticancer agent’s resistance. In the case of the immune checkpoint inhibitors, it seems clear that one of the most important factors allowing for the resistance development remains the PD-1 molecule and its two known ligands: PD-L1 and PD-L2 [23]. Interferon γ plays a pivotal role in this process, enhancing the PD-L1 expression on the tumour cell membranes [13,24,25]. However, on the other hand, the same molecule induces chemokine production, such as CXCL9 and CXCL10, influencing the number of the immune cells infiltrating the tumour, as well as triggering a proapoptotic effect, directly contributing to the programmed death of the cancer cells [24,26,27]. In fact, the majority of tumours resistant to immunotherapy present the impairment of the interferon γ related pathways, in particular those connected with the JAK/STAT signalling pathway. Interestingly, this occurs in both primary and acquired immunotherapy-resistant tumours [27]. In advanced melanoma with acquired immunotherapy resistance (such as anti-PD-1 and anti-CTLA-4 mAbs), loss-of function (LOF) mutations are very frequent, especially in such important genes as *JAK1/2*, *IRF1*, and *IFNGRI1/2* [1,28]. Preliminary results on mouse melanoma models show that it is possible to reverse the immunotherapy resistance simply through JAK/STAT signalling pathway restoration [8,28]. Other experiments indicate that *EGFR* pathogenic variants may impact PD-L1, causing its overexpression on the tumour cells surface. This promotes the development of immunosuppressive microenvironment, just as in the IL-6/JAK/STAT3 signalling pathway aberrations [29]. Besides, increased activation of the MAPK signalling pathway and the inactivating mutations of the *PTEN* gene may interfere with the functioning of T cells and their infiltration inside cancer, namely, tumour infiltrating lymphocytes (TILs) depletion. This is related to the expression of several important factors, such as vascular endothelial growth factor, VEGF, or many anti-inflammatory cytokines, such as IL-8, to name just a few [30,31]. Further research conducted on mice models showed clearly that MAPK inhibitors may restore the proper functioning of TILs, as well as normal interferon γ related pathways and even MHC class I expression, which is often impaired in cancer cells [32,33,34].

Another important factor, appearing to play a part in the immunotherapy resistance, is proteosomal degradation. For example, in healthy cells HIP1R, huntingtin interacting protein 1, tags the PD-L1 molecule for the proteosomal degradation [35]. But inside many cancer cells the *HIP1R* gene has been altered, which in turn causes PD-L1 molecules to be overexpressed and presented in high abundance on the cancer cell surface, leading to the immune cells inhibition, especially T cells anergy [35].

The subsequent gene encoding CD73 protein is also frequently mutated and overly expressed in cancer cells [36,37]. CD73 is an enzyme catalysing dephosphorylation of AMP, leading to the production of adenosine—a molecule crucial for T cells proliferation inhibition, as well as their cytotoxic activities. It has been shown that adenosine present within the tumour microenvironment may also impact angiogenesis and even promote metastasis in several cancer types [36,37,38]. Therefore, high expression of CD73 is related with poor prognosis, for example in the case of pancreatic tumour, as well as some breast cancer subtypes. It is also connected with the resistance to the checkpoint inhibitors, especially anti-PD-1 mAbs, suggesting that CD73 might be considered as a potential biomarker [36,39].

## 4. Factors Related to the Immune System Cells

Cellular pathway abnormalities related to the antigen presentation might prevent T cells from activation and lead to the cancer cell escape from immune system surveillance [40]. Molecular mechanisms usually observed in this category are for example related with the proteasome functioning, antigen processing and transport dysfunctions or structural aberrations of the MHC class I molecules [1]. Mutations and epimutations, as well as transcription inactivation or protein processing malfunctions, or any other processes resulting in erroneous MHC molecule structure, are extremely important, thus their being extensively investigated in the context of immunotherapy resistance [41,42]. The well described loss of function mutation of the β2-microglobulin gene interferes with the splicing, transport and surface expression processes of the MHC molecule, indirectly impacting immunotherapy outcomes [43,44]. Aberrations involving the β2-microglobulin gene have already been well described and tested on different cell lines in vitro, animal models, as well as in patients with advanced melanoma and prostate cancer [40]. Interestingly, some cancers, such as multiple myeloma, may avoid immune system aggression, especially from T cells and NK cells, harnessing mechanisms so complicated like the overexpression of the MHC class I molecules, for example HLA-G [1,45].

There are several already known mechanisms involving T regs that inhibit the activation and proliferation of the T cells, for example using checkpoint molecules such as CD80 and CD86 on the surface of the cells, inhibiting antigen presenting cells, APCs, or blocking the interaction between the T cells and APCs [1,46,47,48]. T regs have the unique ability of killing cancer cells directly, but they may use the same mechanisms to eliminate T cells, APCs or other cells using perforins, granzymes and many cytokines, for example TGF-β [47]. Those mechanisms have already been observed and described in the cancer environment [46,47,48].

Finally, CD80 and CD86 are actually two ligands of the CD28 receptor which plays an important role in immunotherapy resistance. For this reason, there are many ongoing trials focused on the CD28 molecule and potentially combining therapy with the CD28i. In fact, CD28 is a co-stimulatory receptor possessing the ability of activating and enhancing the immunological response, thus it remains an incredibly interesting target for new therapies [23,49].

## 5. Factors Related to the Cancer Microenvironment

Tumour microenvironment, TME, is a very broad term comprising many different molecules and cell types, which form the tumour, together with cancer cells themselves. For example, there are several types of immune cells in the tumour microenvironment, such as T cells, T regs or dendritic cells. Also, there are many cells creating blood vessels within the tumour, and hundreds of regulatory molecules, for example hormones and cytokines. Even different forms of oxygen have been observed within the TME, including reactive forms of oxygen—ROS. The presence or absence of some factors within the TME has already been connected with good prognosis, and/or higher chances of good response to immunotherapy, for example higher amount of TILs is usually a good predictor [22]. Nevertheless, we must remember that the presence of TILs might not be sufficient for a guaranteed good outcome, since T cells are prone to anergy and exhaustion in the TME suppressive condition: although they properly recognise tumour cells as being somehow foreign, they are unable to undertake tumour cells’ elimination nor prevent them from becoming activated [23,50,51].

The tumour microenvironment is a constantly changing place, with many immunological cells infiltrating the tumour being either immunosuppressive, regulatory, or immunomodulatory enhancing. For example, regulatory T cells (T regs), M2 macrophages and myeloid-derived suppressor cells are correlated with the highly immunosuppressive environment, promoting resistance to immunotherapy [6]. Also, cancer-associated fibroblasts (CAFs) with their cytokines produced, as well as angiogenic signals from the stromal cells around the tumour have been shown to affect patient outcomes within the immunotherapy regime [52].

In fact, TME remains a huge therapeutic challenge, especially in the case of brain cancers. Not only because of the blood-brain barrier itself, but also because of the complicated route which activated T cells should undertake in order to act within the tumour, the presence and activity of pericytes, microglia, astrocytes and several other cell types [11]. Their impact on the immunotherapy success or failure still remains unrevealed. Currently, there is no doubt that the specific and highly heterogeneous TME, even if so far unacknowledged, of brain cancer and any brain-located metastases has an enormous impact on immunotherapy’s effectiveness [11,53].

One of the best described immune cells of the TME are tumour-associated macrophages, TAMs. They appear in large numbers within the tumour in response to some chemokines, for example VEGF and GM-CSF [54,55]. At the very beginning of the tumour growth, TAMs are responsible for chronic inflammation, which, despite being a good predictor for immunotherapy, can itself contribute to the carcinogenesis. TAMs can also release ROS forms and many interleukins, for example IL-6 and IL-1β [54]. Moreover, they contribute to the angiogenesis inside developing tumours, promoting further invasion and metastasis, through production of VEGF, EGF and MMP, to name just a few [1,56]. Even more importantly, TAMs are able to inhibit the anticancer response of the T cells, for example through the extensive expression of molecules as: PD-L1, TGF-β, ARGI and PGE2, but also producing chemokine CCL22, which causes accumulation of the T_reg_ cells within the TME, enhancing the immunosuppressive conditions [57,58].

Simultaneous co-expression of many immunological checkpoint molecules on the surface of the tumour cells, as well as other cells present in the TME, together with the generally immunosuppressive circumstances, contribute to the T cells anergy and exhaustion state [23,59,60]. It has been shown that simultaneous co-expression of many immunological checkpoint molecules is not only responsible for the T cells dysfunction but also remains the leading and direct cause of the checkpoint inhibitors’ immunotherapy failure [60]. This is an immediate indicator for the combined immunotherapies, composed of several different checkpoint inhibitors or other therapies, which in turn results in much higher risk of adverse reactions.

Good response to treatment was observed in numerous types of cancer in which combined immunotherapies were administered. This indeed might be a potential way to further improve the efficacies of ICIs [61]. Breast cancer is an example of how different treatments can be effective when united [62]. In vitro and in vivo (mouse model) explorations of synergy between targeted therapies with BRAF and MEK inhibitors, named dabrafenib and trametinib, together with immunomodulatory antibodies targeting PD-1, PD-L1, and CTLA-4 have shown optimistic results in the colon carcinoma cell line CT26 [34].

The combination of various ICIs can overcome some problems of ICI monotherapy in cancers with a certain status [61]. According to results from the CheckMate-227 trial, the first-line treatment with Nivolumab + Ipilimumab resulted in a longer duration of OS than chemotherapy in NSCLC patients [63]. Additionally, in the CheckMate-067 trial, patients with metastatic melanoma receiving Ipilimumab + Nivolumab therapy for the first time had a better ORR and 5-year survival rate compared to Nivolumab or Ipilimumab alone [64].

Nevertheless, previously used therapies may have a tremendous impact on the TME. In fact, we can modulate TME conditions using other forms of therapy, for example traditional chemotherapy or BRAF or MAPK inhibitors, leading to the TILs depletion and lowering chances of further immunotherapy success [65]. Influx of the cells and substances would not be possible without properly elaborated blood vessels within the tumour. High levels of VEGF produced by the cancer cells or injected during the therapy regimen can also be immunosuppressive, suspending dendritic cells’ maturation through NFkB pathway activation [66]. Tumour vascularisation is in fact dysfunctional and structurally impaired, and tumour-related endothelial cells show low expression levels of adhesion molecules, further hindering the penetration of the cancerous tissue by the immune system cells [66,67]. Abnormal vascularisation directly contributes to the metastasing potential, but also leads to the hypoxic conditions inside some tumour areas, resulting in necrosis progression [66].

## 6. Factors Emerging from the Host Cells—Cancer Cells Interactions

The presence of the neoantigens may be seen as a positive factor, allowing for personalised medicine to play its part, for example, through personalised cancer vaccines. On the other hand, neoantigens may be seen as a negative factor, especially in the long run: immune cells recognise cancer cells with the neoantigens and actively eliminate them, whereas other clones may also be still present. In this scenario, a patient’s organism may indirectly promote the development of a tumour invisible to the immune system. Its cells may have no neoantigens visible to the immune system, and thus are impossible to be eliminated [1,68]. This phenomenon is known as the escape of the tumour cells from immune surveillance, however the mechanism described above is just an example of several possible ways of an immune escape [1,68].

Not surprisingly, several germline mutations have been also connected with the cancer response to immunotherapy, especially inside genes involved in DNA repair mechanisms or checkpoint proteins, but also the INFγ signalling pathway, antigen presentation mechanisms and immune response itself [69]. For example, it has been shown that overall survival of patients with the germline variant of the gene *PDCD1* 804C>T (rs2227981) decreased significantly: for the wild-type variant the three-year survival rate was 71%, whereas for the mutant variant the three-year survival rate was 51.8% [6,70]. In fact, this single nucleotide genetic variant affected the clinical efficacy of immune checkpoint inhibitors, reducing the initiation of the transcription, as well as the expression of the PD-1 protein on the T cell surface [70]. A different single nucleotide variant, *ALDH2* rs671, playing a key role in the detoxification of endogenous acetaldehyde, resulted in decreased enzyme activity [71]. Studies have shown that this variant may enhance the presentation of tumour antigens, especially those caused by the DNA damage triggered by acetaldehyde [6,71]. More importantly, the same variant was also responsible for inhibiting peripheral blood T cell amounts and the activation of T cells, and in thoracic cancer it was shown to be a negative predictor of the immune checkpoint inhibitors’ therapy effectiveness [71]. In particular, the last property has been investigated extensively, leading to the conclusion that the *ALDH2* rs671 variant may have inhibited the PI3K -AKT pathway in the T cells, resulting in the accumulation of the endogenous aldehydes that in turn negatively affected the immune checkpoint inhibitors efficacy in thoracic cancer patients [71].

Another important mechanism observed in the TME significantly impacting the interactions between the immune cells and tumour cells is tryptophane catabolism [72]. Tryptophane is a key molecule essential in the process of T cell activation. There are several enzymes inside the immune synapse playing a part in tryptophane metabolism, such as IDO or TDO, being actively produced by many cancer cells. Those enzymes convert tryptophane to kynurein and other metaboliltes, useless for the T cells [72,73]. As a result, T cells cannot be activated in an environment without tryptophane. Thus, tryptophane-depleted TME causes T cells dysfunction leading to their anergy and exhaustion, sometimes even directly to their apoptosis [72]. In order to check if it is possible to counteract this process, an interesting experiment has been conducted on a mice melanoma model: anti-CTLA-4 antibodies have been administered together with an IDO enzyme inhibitor (1-metylotryptophan, 1MT) [73]. The results were very encouraging, showing that tumour cells’ sensitivity to the treatment used was clearly enhanced, suggesting that IDO inhibitors may become an important immunotherapy adjuvant method in the near future [73].

Similarly, cholesterol metabolites produced by the tumour cells, such as hydroxycholesterol, might activate hepatic pathways related with immunosuppression, impair proliferative ability of the lymphocytes and dendritic cells, as well as stimulate maturation and differentiation of the T_h17_ cells [74,75]. In several studies conducted on the animal models of melanoma and pancreatic cancer, it has been shown that abrogating cholesterol esterification suppresses the growth and metastasis of cancer. Those adjuvants have been added to the regular immunotherapy regimen, effectively enhancing CD8+ T cells actions and increasing treatment success rates [76,77]. It is clear, however, that the presence of the CD8+ T lymphocytes in the TME is a positive predictor of a favourable treatment outcome, at least increasing chances of the possibly better response to immunotherapy, including checkpoint inhibitors [78].

Moreover, many tumours can release exosomes into the circulation, encapsulating such important molecules as PD-L1 inhibitory protein [7]. Extracellular vesicles known as exosomes contain DNA, RNA and proteins of the cells that secrete them [79]. They are usually taken up by the distant cells and this way their content may affect cell function and behaviour [79]. In the context of immunotherapy, it has been shown that exosomal PD-L1 coming from the tumour cells may inhibit the activation of the CD8+ T cells and even play a role in the tumour lymphatic metastasis process [7,80,81,82]. Moreover, studies in mice have shown that exosomes of the bone-marrow derived cells (BMDCs) can also contain PD-L1 capable of inhibiting the proliferation and activation of the CD8+ T cells, thus playing an important role in tumour suppression [83]. In fact, this crucial information may explain why some patients whose tumour cells do not express the PD-L1 molecule actually do respond to immunotherapy: anti-PD-L1 drugs can remove immunosuppression caused by exosomal PD-L1, restoring antitumour immunity [83]. The above-mentioned actions may induce apoptosis among T cells, including cytotoxic T cells present in the TME [1,7]. What is more, regulatory T cells can be stimulated, both those which are present in the TME and in the circulatory system [7]. All those actions contribute to the suppression of proper immunological response and the facilitation of tumour cells escape, leading to the acquired immunotherapy resistance.

## 7. Other Important Factors

Latest reports also suggest several other factors potentially participating in the immunotherapy resistance, however further research is clearly required to clarify those assumptions, legitimating and explaining them on the molecular level. Unsurprisingly, some of those factors are rather common, such as advanced age, biological sex, diet, many hormones, existing comorbidities, drugs, and the gut microbiome [1,84,85].

Ageing is inseparably connected with the impaired functioning of the immune system, therefore there is an intuitive suspicion that elderly people may respond worse to the immunotherapy. Indeed, some cases have been described that confirm this notion. However, no significant difference has been noted in the safety nor effectiveness profiles of the immune checkpoint inhibitors usage between younger and older cancer patients [86]. Interestingly, the differences between response among male and female patients are statistically significant, not solely in one single study, but even if combined in a large meta-analysis comprising over 11,000 patients with advanced cancer. Better immunotherapy response and overall survival rate was observed among men, at 67%, than women, at 33% [87]. This result clearly confirms the notion that hormones have a strong say with regard to immunotherapy treatment [88].

One of the most important factors contributing to immunotherapy success is body mass index, which has been confirmed not only on mice studies, but also in cancer patients themselves. Astonishing enough, it seems like obesity may be a favourable factor in the immunotherapy regimen. The treatment of obese mice with anti-PD-1 antibodies was more successful and connected with much less adverse reactions [88]. Finally, the very surprising impact of the gut microbiome, still being intensively explored, seems to correlate with the immunotherapy response: the outcome is more favourable if there are certain bacterial strains, such as *Bacterioides thetaiotaomicron* or *Bacterioides fragilis*, even if the mechanism of their actions remains totally mysterious [89,90,91]. Lately, it has been shown that the faecal microbiome transplant containing those strains can increase immunotherapy susceptibility among those individuals that were so far resistant: anti-CTLA-4 antibodies have been tested on mice models, with an amazingly good outcome [91].

Some sources claim that antibiotics use may be an independent risk factor of the immune checkpoint inhibitors resistance, contributing to lower overall survival and progression-free survival rates, as well as higher risk of cancer progression [84,92,93,94]. Also, proton pump inhibitors, drugs reducing the amount of stomach acid produced by the glands lining the stomach walls, may cause immunosuppression, as they reduce the expression of adhesion molecules of the cells within the inflammatory region, or through the modulation of secreted proinflammatory cytokines [93,94,95]. There are several other drug types that may affect the effectiveness of immune checkpoint inhibitors, such as opioids, although further studies are required to confirm their direct impact and exclude concomitant factors, such as the higher prevalence of alcohol consumption or the usually lower body mass index among people taking opioids [94].

## 8. TME-Based Classification of Tumours

Recent multi-omics studies divided tumours into several different types, according to the components of their microenvironment, especially unique vascular, stromal and cytokine expression patterns:

IE/F—immune-enriched, fibrotic tumour; high abundance of the functional gene expression signatures related to angiogenesis and activation of cancer-associated fibroblasts.

IE—immune-enriched, nonfibrotic; characterized by the high degree of immune infiltration and high cytokine concentration, high tumour mutation burden score (TMB), high CD8+ T cells to T regs ratio, high M1/M2 macrophages ratio, and increased activation of the JAK/STAT pathway.

F—fibrotic; little or no leukocytes, high abundance of the functional gene expression signatures related to angiogenesis and activation of cancer-associated fibroblasts.

D—desert/depletion; little or no immune cells present, highest percentage of malignant cells [6,52].

Interestingly, these subtypes were conserved across at least 20 different cancer types analysed in the study among over 8500 samples included in the experiment, showing that many biological processes and immunological activities are similar irrespective of the cancer type [52]. Even though cancer cells are unique, and they clearly possess different genetic abnormalities, it seems like immune relationships may be similar in particular TME subtypes [52].

Significantly higher overall response to the checkpoint inhibitors has been noted among patients with tumour microenvironment subtype IE, and prolonged progression-free survival and overall survival have also been observed among subtype IE [52]. Among specific cancer types, the immunosuppressive TME subtype F in bladder cancer patients presented lower response rates to immunotherapy and have a generally worse prognosis, compared with the other TME subtypes [52]. In the case of melanoma, those patients possessing TME subtypes F and D experienced the worst outcomes to the immune checkpoint inhibitors, whereas IE TME carriers have generally favourable prognosis and may benefit the most from the immunotherapy [52].

Putting this tissue-independent TME classification into broader perspective, it may challenge the idea of cancer immunogram or cancer-related set-points, adding another brick to the wall of the concept of personalised medicine [52,96,97]. Classification based on the TME subtypes rather than tissue-of-origin can further stratify patients and predict their immunotherapy response, especially when combined with genomic data and other omics results.

## 9. Immunotherapy-Related Adverse Events

Undoubtedly, treatment-related adverse events of the checkpoint inhibitors are abundant, especially if administered in combinations [98,99]. The most commonly reported adverse events were anaemia (45.4%), fatigue (34.3%) and dysphagia (30%), although among grade 3 or higher adverse events, the most common were neutropenia (19.6%), hypertension (9.3%) and lymphopenia (10.3%), but also pyrexia, diarrhoea, skin toxicity, haematological events, thyroid dysfunction, endocrinological aberrations and pneumonitis, to name just a few [100]. Treatment related death is rather rare, although not impossible, and in most cases, it is caused by the severe toxicity of the combination treatment or the aggressive reaction from the immune system, including cytokine storm. Especially among patients receiving both chemotherapy and anti-PD-1 or anti-PD-L1 inhibitors, grade 3 or higher adverse events were very frequent and often severe and prolonged [98,100]. Thus, balancing the efficacy and safety of the regimens should be accurately considered with regard to the individual cases. Special attention should be paid with regard to the patients receiving combination therapies including anti-CTLA-4 therapies: it has been shown that such combinations may trigger hypophysitis more frequently due to the ectopic expression of the CTLA-4 molecule in the pituitary gland cells, leading to direct damage that is difficult to treat if unnoticed at the initiation [101]. Moreover, combinations comprising both immunotherapy and targeted therapies such as VEGF or VEGFR inhibitors are more likely to cause hypertension and severe proteinuria, rare adverse events attributed to the anti-angiogenic effects of those drugs [102,103]. Generally, chemotherapy combinations are clearly associated with a higher risk of any adverse events, compared with other combination treatments comprising immunotherapy.

A recently published extensive meta-analysis disclosed an incidence of treatment-related adverse events of as much as 97.7% in the chemotherapy-immunotherapy combination regime [100]. Notably, for grade 3 or higher adverse events, as many as 68.3% of cases have been reported [100]. In the targeted therapy with immunotherapy combinations, 94.5% of cases with adverse events have been noted for all grade adverse events and as many as 47.3% for grade 3 or higher [100]. For the radiotherapy-immunotherapy combinations, 89.4% of cases with all-grade adverse reactions have been observed, including 12.4% for grade 3 or higher [100]. In case of the immunotherapy combinations, 86.8% of cases with adverse reactions have been reported, among which 35.9% were for grade 3 or higher [100].

The above-mentioned adverse events have been presented in Table 2.

## 10. Future Perspectives of the Immune Checkpoint Inhibitor Therapy

It is believed that ICIs and chimeric antigen receptor (CAR)-T cells have the greatest potential to further improve the prognosis for cancer patients, with ICIs showing better clinical results than CAR-T cells in treating solid tumours [61].

First of all, the key to an effective treatment is, as often said, “teamwork” (combining different existing therapies), for example immunotherapy in addition to adjuvant and neoadjuvant protocols in breast cancer [62]. Despite the magnificent and highly encouraging performance of checkpoint inhibitor therapies in some cases, in other cancer types or patient groups, response rates remain poor, indicating the need for combination regimes to be developed and optimised to synergistically reverse the tumour immunosuppressive microenvironment [104,105,106].There are many ongoing trials combining different immunotherapy drugs or immunotherapy plus chemotherapy, radiotherapy or even targeted therapies. For example, significantly longer OS and PFS have been achieved compared with standard chemotherapy, in randomised clinical trials leading to the approval of anti-CTLA-4 with anti-PD-1 antibodies together in combination for the treatment of renal cell carcinoma, advanced melanoma and colorectal cancer [107,108,109]. A very interesting combination of the angiogenesis inhibitor with the anti-PD-1 antibody has been successfully approved for renal cell carcinoma [110]. In the future, the development of combination immunotherapies can make breast cancers and immunologically cold lesions, become immune-activated tumours ready for response to immunotherapy [62]. In order to boost a weak antitumour immunity, it is crucial to investigate the mechanisms responsible for hot, altered or cold immune tumours [111].

Another exciting prospect in the near future of immunotherapy is the development of vaccines [62]. For example, in prostate cancer a prolonged survival was observed in men with asymptomatic or minimally symptomatic metastatic CRPC [112]. Promising effects of dendritic cells (DCs) vaccination with GP96 and SMP30 were seen in vitro for hepatoma (more potent antitumor effects because of GP96 and SMP30 stimulation of DCs) [113].

Moreover, it is necessary to identify environmental modifiers of immunity (the microbiome, metabolic and hormonal parameters, and concurrent drug therapy) as well as to keep looking for new biomarkers that could predict the response and resistance to therapy in order to select proper patients to receive ICIs [61,62].

Another challenge for immunotherapy that should be meticulously investigated in the future are the irAEs. Although various mechanisms have been proposed to explain the development of irAEs, the exact pathophysiology still needs to be explored [61,114]. A few major cancer societies have elaborated recommendations on how to manage irAEs in order to provide clear and consistent references regarding the immunotherapy toxicity [114].

In order to make immunotherapies more effective it is critical to understand the mechanisms of resistance, both primary and acquired [1]. Primary resistance mechanisms involve tumour cell-extrinsic factors (that involve components other than tumor cells within the TME) and tumour cell-intrinsic factors (such as expression or repression of genes in cancer cells that prevent infiltration or activation of immune cells within the TME) [61].

Also, pharmacodynamics and pharmacokinetics of the drug classes used should be considered, since there are known differences between checkpoint inhibitors already available on the market [115,116]. To give an example, pembrolizumab (anti-PD-1 antibody) seems to have better engagement and affinity among all PD-1 inhibitors, whereas avelumab (anti-PD-L1 antibody) has the best affinity among PD-L1 inhibitors, and atezolizumab (anti-PD-L1 antibody) is known to have the longest half-life among PD-L1 inhibitors [115,116].

What remains a great challenge for immunotherapy is finding treatment for aggressive malignancies with limited treatment options [117,118]. For example, in biliary tract cancer (BTC), most patients show disappointing clinical outcomes and only a small group of BTC patients respond well to immune checkpoint inhibitor therapy and, according to available data, it is currently not possible to determine which biomarker makes patients more likely to respond well to ICIs [117]. Another such challenge is the hepatocellular carcinoma (HCC), where most patients do not respond to immunotherapy, and only a small percentage of them achieve durable responses. Again, there’s no single biomarker that can select HCC patients who are likely to respond well to immunotherapy [118].

## 11. Summary

Summing up, it is important to mention hyperprogression as the last phenomenon regarding immunotherapy which also remains abstruse, yet very important and typical to the treatment with the use of immune checkpoint inhibitors [119,120]. It is not a typical form of the immunotherapy response; in some studies it is described as a special form of it, even necessary in some cases. Hyperprogression is a phenomenon of unexpectedly rapid disease progression in response to the immunotherapy drug administration, much faster than it probably would have progressed with no medication at all. Although the background of this mysterious state remains unrevealed, there are some genetic hints for further research. In mice, *Mdm2/4* amplifications, several variants of the *EGFR* gene, as well as structural aberrations of the 11 chromosome seem to play an important role in the occurrence of hyperprogression [119,120]. Nevertheless, there are no confirmed molecular defects known among patients with hyperprogression. Existing studies comprise only a small number of participants, and in vitro or mice studies should not be directly extrapolated into human beings.

Therefore, we are more and more aware of the multitude of factors contributing to the development of immunotherapy resistance, both in primary and acquired forms. The accumulating number of upright studies comprising a huge number of participants involved in the studies worldwide still support the argument in favour of immunotherapy, with more arguments supporting its miraculous actions, even if we are aware of its limitations and the constant need of improvement with further research.

## Figures and Tables

**Figure 1 ijms-23-02847-f001:**
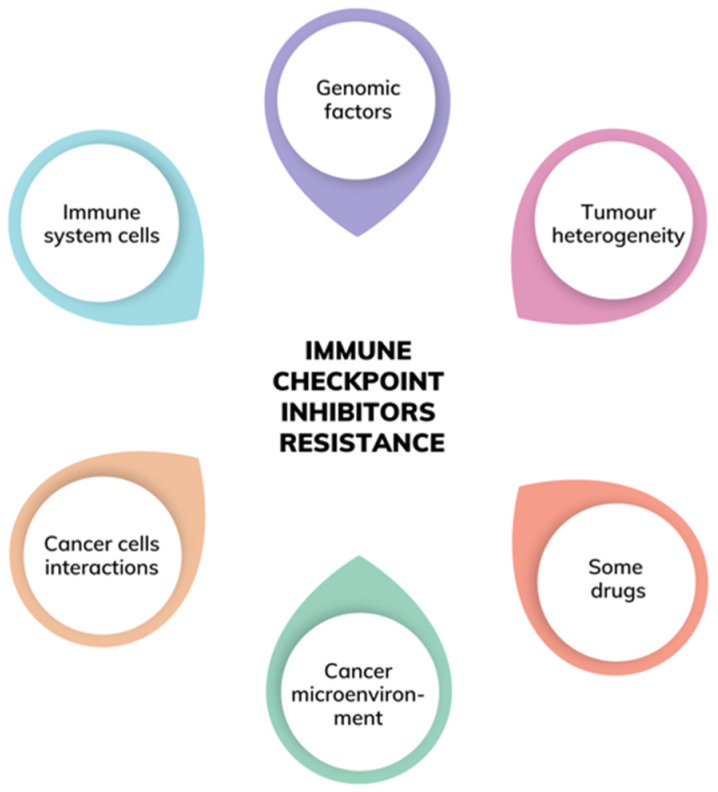
Factors influencing the immunotherapy resistance [6].

**Table 1 ijms-23-02847-t001:** Primary versus acquired resistance to immunotherapy.

	Primary	Acquired
DEFINITION	resistance occurs when cancer does not respond to the treatment which has never been used in this patient before [1,2,3]	resistance occurs when the drug used before no longer works, despite being successfully used in the past in particular patients [1,2,3]
CRITERIA	exposure to the drug for at least six weeks, stable disease or progressive disease as a response for at least six months, confirmatory scan for progressive disease after at least four weeks of the initial progression [6]	at least six months of progression-free survival must be observed to meet the criteria of acquired resistance [1,2,3]
MECHANISM	immune system does not take any actions despite the pharmacological stimulation used [2,3]	suppression of proper immunological response and the facilitation of tumour cells escape [1,7]
FACTORS	lack of antigen expression, constitutive PD-L1 expression or other co-inhibitory molecules, EGF/EGFR expression abnormalities, loss of HLA expression or presentation, improper activation of the MAPK pathway, WNT/β-catenin pathway activation, loss of the PTEN expression leading to the enhancement of PI3K pathway, abrupted interferon pathway signalling, especially INFγ, JAK/STAT signalling pathway abnormalities [6]	
EXAMPLE	mutations in tyrosine–protein phosphatase non-receptor type 2 (Ptpn2) have been associated with primary resistance to PD-1 blockade via resistance to IFN-γ [8]	among melanoma patients treated with immune checkpoint inhibitors, about 30% of them do respond well at the beginning of the treatment but develop acquired resistance during the treatment regime up to the point they finally stop responding to this form of immunotherapy [4,5]

**Table 2 ijms-23-02847-t002:** Most commonly reported treatment-related adverse events [98,100].

Adverse Event	Frequency	References
anaemia	(45.4%),	[100]
fatigue	(34.3%)	[100]
dysphagia	(30%)	[100]
neutropenia	(19.6%)	[100]
hypertension	(9.3%)	[100]
lymphopenia	(10.3%)	[100]
death	rare	[98,100]

## Data Availability

Not applicable.

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
