# Peer review of "Challenges of the Immunotherapy: Perspectives and Limitations of the Immune Checkpoint Inhibitor Treatment"

_ijms, 2022, doi:10.3390/ijms23052847_

Round 1
Reviewer 1 Report
The work submitted for review in my opinion fully deserves publication. It concerns very current issues related to immunotherapy. The work is written in an understandable and legible way, it presents the current state of scientific knowledge, the work is based on the latest publications from the last few years. I rate the manuscript very highly. I am not submitting any corrections to the text of the article.
Yours sincerely
Author Response
Dear Reviewer,
We would like to thank you for your careful and thorough reading of this manuscript and your constructive suggestions, which undoubtedly help to improve the quality of this manuscript.
We decided to change the main body slightly and improve the conclusions, but also add several tables and pictures for the clarity.
Once again, thank you for your time and helpful insights,
Sincerely yours,
The authors
Reviewer 2 Report
Dear Editor, thank you so much for inviting me to revise this manuscript.
This study addresses a current topic.
The manuscript is quite well written and organized. English could be improved.
Figures and tables are comprehensive and clear.
The introduction explains in a clear and coherent manner the background of this study.
We suggest the following modifications:
- Introduction section: although the authors correctly included important papers in this setting, we believe some studies regarding biomarkers of response to immunotherapy should be cited within the introduction (PMID: 33535621 ; PMID: 33549983 ), only for a matter of consistency. We think it might be useful to introduce the topic of this interesting study.
- The authors should expand some sections, including a more personal perspective to reflect on. For example, they could answer the following questions – in order to facilitate the understanding of this complex topic to readers: What are the knowledge gaps and how do researchers tackle them? How do you see this area unfolding in the next 5 years? We think it would be extremely interesting for the readers.
However, we think the authors should be acknowledged for their work.
We believe this article is suitable for publication in the journal although some revisions are needed. The main strengths of this paper are that it addresses an interesting and very timely question and provides a clear answer, with some limitations.
We suggest a linguistic revision and the addition of some references for a matter of consistency. Moreover, the authors should better clarify some points.
Author Response
Dear Reviewer,
We would like to thank you for your careful and thorough reading of this manuscript and for all the comments and constructive suggestions, which undoubtedly help to improve the quality of this manuscript. We’ve been happy to implement all the suggestions, including language correction, and correct the typos and omissions.
Thus, we are grateful for the suggestion about the missing papers (https://pubmed.ncbi.nlm.nih.gov/33549983/ and https://pubmed.ncbi.nlm.nih.gov/33535621/ ), they are included in the current version. Furthermore, we decided to change the main body slightly and improve the conclusions, but also add several tables and pictures for the clarity. As you’ve mentioned, we hope that immunotherapy remains a game-changer in the field, transforming the way we treat cancer.
Once again, thank you for your time and helpful insights,
Sincerely yours,
The authors
Round 2
Reviewer 2 Report
The authors modified the manuscript according to our suggestions.
We recommend Acceptance.